CD161, a promising prognostic biomarker in hepatocellular carcinoma, correlates with immune infiltration

Wang Jinfeng 1
Wang Xiaoxiao 2
Shi Jianfei 1
Wang Yongjun 2
Mi Lili 1
Zhao Man 1
Han Guangjie 1
Yin Fei 47100214@hebmu.edu.cn 1
1 Department of Gastroenterology, The Fourth Hospital of Hebei Medical University , Shijiazhuang , China
2 Department of Pathology, The Fourth Hospital of Hebei Medical University , Shijiazhuang , China
Uversky Vladimir
Electronic publication date: 2025 Mar 17
Publication date: 2025
Volume: 13
Electronic Location ID: e19055
Received 2024 Sep 26; Accepted 2025 Feb 5
Copyright: ©2025 Wang et al.
Copyright year: 2025
Copyright holder: Wang et al.
License: This is an open access article distributed under the terms of the Creative Commons Attribution License, which permits unrestricted use, distribution, reproduction and adaptation in any medium and for any purpose provided that it is properly attributed. For attribution, the original author(s), title, publication source (PeerJ) and either DOI or URL of the article must be cited.
License URL: https://creativecommons.org/licenses/by/4.0/

Keywords: CD161, Prognostic biomarker, Immune infiltration, Tumor immune microenvironment, Immunotherapy, Hepatocellular carcinoma

Funding: Hebei Natural Science Foundation H2020206589 Medical Science Research Project of Hebei No. 20240474 This work was supported by Hebei Natural Science Foundation (H2020206589) and the Medical Science Research Project of Hebei (Grant No. 20240474). The funders had no role in study design, data collection and analysis, decision to publish, or preparation of the manuscript.

==============================
Background

CD161, encoded by the killer cell lectin-like receptor B1 (KLRB1) gene, exhibits varied roles among different tumors. This study aimed to explore both the potential value of CD161 as a prognostic biomarker for hepatocellular carcinoma (HCC) and its association with immune cell infiltration.

Methods

A total of 109 HCC patients who underwent surgery were retrospectively analyzed. Immunohistochemistry, bioinformatic analyses, and statistical measurements were used to investigate the associations between CD161 expression, immune cell infiltration, and clinical outcomes in both public databases and in-house cohorts.

Results

CD161 was highly expressed at both protein and mRNA levels in adjacent normal tissues compared to tumor tissues of HCC patients. Meanwhile, CD161 was enriched in HCC cases characterized by smaller tumor sizes (≤5 cm) and the absence of portal vein tumor thrombus. Individuals with high CD161 expression showed extended overall survival (OS) and relapse free survival (RFS) compared to those with lower CD161 levels. CD161 was identified as an independent prognostic indicator for both OS and RFS. In addition, the enrichment analysis indicated a close correlation between CD161 and immune response, as well as between CD161 and the signaling pathways of cytokines and chemokines, implying its role in immune regulation during cancer development. Specifically, CD161 expression was positively associated with immunomodulators and tumor-infiltrating immune cells, especially CD8+T cells, CD4+T cells, and dendritic cells. Multiple public databases showed that patients with high CD161 expression were more likely to derive benefits from immunotherapy.

Conclusion

CD161 was identified as a promising prognostic biomarker for HCC, as its expression indicates a favorable prognosis. Additionally, CD161 is closely linked to high infiltration of immune cells, participates in the regulation of the tumor immune microenvironment, and holds promise as a potential biomarker for predicting the efficacy of immunotherapy.

Introduction

Hepatocellular carcinoma (HCC) continues to be a major global public health concern, with rising incidence and mortality rates (McGlynn, Petrick & El-Serag, 2021; Sung et al., 2021). Despite advancements have been made in early detection, a large proportion of HCC patients are still diagnosed at an advanced stage, leading to a poor prognosis (Llovet et al., 2016). Consequently, there is an urgent need to explore new biomarkers to predict HCC prognosis and facilitate the selection of optimal therapeutic approaches.

The tumor immune microenvironment (TIME), which involves a complex interaction between tumor cells, immune cells, and cytokines, has been demonstrated to be closely associated with tumor development, recurrence, and metastasis (Lv et al., 2022). The infiltration of immune cells in the tumor microenvironment is critically important for the progression and prognosis of tumors. Tumor-infiltrating immune cells play a dual role in HCC: They are capable of effective cancer cell clearance and induce antitumor immune responses; however, they also promote tumor progression by causing immune dysfunction (Llovet et al., 2022). Immune checkpoints located on tumor cells are potent tools that facilitate unrestricted tumor cell growth and evasion of host immunosurveillance in the tumor immune microenvironment. Advancements in immunotherapy have led to the creation of humanized monoclonal antibodies designed to block immune checkpoints like PD-1, CTLA-4, and LAG3, et al. In recent years, immune checkpoint inhibitors have revolutionized the management of HCC. The combination of the immune checkpoint inhibitor atezolizumab and the humanized monoclonal antibody bevacizumab outperformed the kinase inhibitor sorafenib in terms of survival and adverse events, and has emerged as a new first-line therapy in advanced hepatocellular carcinoma (Finn et al., 2020). Nevertheless, only approximately one-third of HCC patients can benefit from immunotherapy. The tumor immune microenvironment plays a pivotal role in determining the efficacy of immunotherapy. Multiple studies have indicated that immune cells, particularly tumor-infiltrating lymphocytes, play a significant role in predicting the efficacy of immune checkpoints inhibitors (Llovet et al., 2022; Holder et al., 2024). Therefore, identifying biomarkers that regulate the TIME is crucial for enhancing antitumor immunity, improving prognostic outcomes, and selecting the optimal treatment for patients with HCC.

CD161, which is encoded by killer cell lectin-like receptor subfamily B member 1 (KLRB1), has recently been proposed as a potential inhibitory receptor and a promising novel target for immunotherapeutic treatments of glioma (Mathewson et al., 2021). Additional research revealed that CD161 promotes tumor progression due to its unique effect on T cell dysfunction in glioma (Di et al., 2022). Moreover, Lao et al. (2023) discovered that CD161-overexpressing CD8+ T cells were enriched in chemo-resistant tumors. The presence of CD161 expression in CD8+ T cells was correlated with chemoresistance and shortened patient survival. Conversely, prior research has demonstrated that CD161 expression is potentially linked to improved clinical outcomes in lung cancer (Braud et al., 2018), breast cancer (Weng et al., 2022; Huang et al., 2023), oropharyngeal cancer (Wei et al., 2023), and endometrial cancer (Liang et al., 2023). Additionally, numerous studies have demonstrated a close association between CD161 and the tumor immune microenvironment, highlighting its significant role in immune regulation (Li et al., 2022; Hu et al., 2024). However, the significance of CD161 expression in HCC tissue, its potential role in the prognosis of HCC, and its relationship with immune cell infiltration are still unclear.

The present study compared the expression of CD161 in HCC and adjacent tissues and investigated the correlation between CD161 expression and various clinicopathological characteristics. Additionally, the clinical significance of CD161 in predicting overall survival (OS) and recurrence-free survival (RFS) in HCC patients was assessed. The biological functions of CD161 and their associations with immune cell infiltration were also examined. Lastly, the correlation between CD161 expression and treatment response was evaluated in two cohorts undergoing immunotherapy. This study aimed to establish the prognostic value of CD161 in HCC patients, elucidate its relationship with immune cell infiltration, and assess its potential as a biomarker for immunotherapy.

Materials & Methods

Patients and specimens

Clinical data and surgical specimens were retrospectively collected from 109 patients diagnosed with HCC at the Fourth Hospital of Hebei Medical University between November 2017 and January 2023. All patients had undergone curative surgical resection, and the presence of hepatocellular carcinoma was confirmed by postoperative pathology. A range of clinical and pathological indicators were gathered, including age, gender, history of hepatitis, tumor size, tumor number, albumin-bilirubin (ALBI) score, alpha-fetoprotein (AFP) levels, neutrophil-to-lymphocyte ratio (NLR), platelet-to-lymphocyte ratio (PLR), tumor-node-metastasis (TNM) staging, China National Liver Cancer (CNLC) stage (Xie et al., 2023), Barcelona Clinic Liver Cancer (BCLC) staging (Reig et al., 2022), portal vein tumor thrombus (PVTT) occurrence, and microvascular invasion (MVI) status. The study received approval from the ethics committee of the Fourth Hospital of Hebei Medical University (certificate no. 2024KS003), and written informed consent was obtained from all participants.

Immunohistochemistry (IHC)

The expression levels of CD161 in HCC and adjacent normal tissuestissues were assessed by immunohistochemistry analysis. Following standard IHC procedures, formalin-fixed paraffin-embedded (FFPE) samples were blocked with 10% goat serum (cat. no. ZLI-9056; ZSGB-BIO Technologies) for 10 min at 37 °C prior to incubation with primary antibodies against CD161 (1:500, HA601063, HUABIO) at 4 °C overnight. The negative controls were prepared using phosphate-buffered saline instead of the primary antibody. Positive controls were generated according to the manufacturer’s instructions provided with the antibody. A goat anti-rabbit secondary antibody was then applied at room temperature for 20 min. Lastly, hematoxylin was used to counterstain nuclei. The expression levels of CD161 were observed under a light microscope at x200 magnification with five randomly selected fields of view for a separate semi-quantitative analysis, and the average was calculated as the final immunohistochemical score. The immunoreactivity score was blindly evaluated by two senior pathologists. The immunoreactivity score (IRS) of CD161 was calculated using the proportion of positive cells (PP) and the staining intensity (SI), IRS = SI*PP (Zhou et al., 2022). For SI scoring, colorless scored 0 points, pale yellow scored 1 point, brown scored 2 points, and tan scored 3 points. For PP scoring: 0 for negative cells, 1 for positive cells with a 10% chance of being positive, 2 for 11% with a 50% chance of being positive, 3 for 51% with a 100% chance of being positive. The median immunoreactivity score served as the cut-off value to define high and low expression in the samples.

Data acquisition

In this study, the mRNA expression profiles and clinical features of HCC patients (n = 365) with complete survival data were collected from The Cancer Genome Atlas (TCGA) database (https://cancergenome.nih.gov/). The processed microarray data of 221 HCC samples from GSE14520 were retrieved from the Gene Expression Omnibus (GEO) database (https://www.ncbi.nlm.nih.gov/geo/) to confirm the relationship between CD161 expression and clinicopathological features and HCC survival. To confirm the biological function of CD161 expression in HCC, another dataset (ICGC-LIRI-JP) including the mRNA expression profiles of 230 patients was obtained from ICGC (https://dcc.icgc.org/; The ICGC Data Portal officially closed in June 2024. The related data is now available at https://www.icgc-argo.org). The TCGA, ICGC, and GSE14520 contain both core tumor tissue and adjacent tissue. Both cancer tissues and adjacent normal tissues were used to examine the differential expression of CD161 and its diagnostic significant for HCC. All other analyses used exclusively core cancer tissue. The mRNA sequencing data of the TCGA-LIHC and ICGC-LIRI-JP cohorts were converted into transcripts per kilobase million (TPM) values and were then log2(TPM+1) transformed, which has been recommended as the most accurate quantification method with minimal statistical biases (Jin, Wan & Liu, 2017). An immunotherapy cohort (GSE140901) containing 24 HCC patients was acquired from the GEO database. The IMvigor210 cohort (Mariathasan et al., 2018), a study of atezolizumab in patients with locally advanced or metastatic urothelial carcinoma, was downloaded from http://research-pub.gene.com/IMvigor210CoreBiologies. The protein levels of CD161 in human normal liver tissue and HCC were examined using the Human Protein Atlas (HPA: https://www.proteinatlas.org/) database.

Prognostic analysis

Kaplan–Meier analyses were performed to evaluate the overall survival and recurrence-free survival in both TCGA and GSE14520 datasets. Furthermore, the prognostic significance of various parameters was assessed using univariate and multivariate Cox proportional hazards models.

Functional enrichment analysis

The genes most relevant to CD161 were uploaded to the Database for Annotation, Visualization, and Integrated Discovery (DAVID). The enrichment results of Gene Ontology (GO) analysis and Kyoto Encyclopedia of Genes and Genomes (KEGG) pathway analysis were retrieved. The figures displayed the top six results in ascending order of P-value (P < 0.05).

Gene set variation analysis (GSVA)

The gene list of immune processes was downloaded from the Gene Set Enrichment Analyses database (GSEA, https://www.gsea-msigdb.org/gsea/index.jsp). An enrichment score was calculated for each HCC sample using the provided package in the R environment. A heatmap was generated based on the enrichment results and Pearson’s correlation coefficients were computed.

Immune cell infiltration analysis

The immune cell infiltration score in both TCGA and ICGC databases underwent xCell deconvolution analysis, CIBERSORTx (https://cibersortx.stanford.edu/; Newman et al., 2015), and MCP-counter analysis. The levels of immune cell infiltration were compared between different groups, which were stratified based on the median CD161 expression level in the TCGA and ICGC databases. To further elucidate the relationship between CD161 and immunophenotyping, data was used from a published work (Thorsson et al., 2018), which identified six immune subtypes (Wound Healing, IFN-γ Dominant, Inflammatory, Lymphocyte Depleted, Immunologically Quiet, and TGF-β Dominant) based on the TCGA database. The leukocyte fraction, lymphocyte infiltration signature score, B-cell receptor (BCR) Shannon, and T-cell receptor (TCR) Shannon were compared across different CD161 groups.

Statistical analyses

All statistical analyses and visualization were performed using SPSS 25.0 (IBM Corp), R software (version 4.2.1), Python (version 3.9) and Prism 10.0 (GraphPad, San Diego, CA, USA). Unpaired Student t-tests or Mann–Whitney U tests (also known as the Wilcoxon rank-sum test) were used to compare the differences between the two groups. One-way ANOVA or Kruskal–Wallis tests were used for comparisons of more than two groups. The associations between CD161 expression and the patients’ clinicopathological characteristics and treatment efficacy were evaluated by the chi-square test or Fisher’s exact test. The receiver operating characteristic (ROC) curve was generated using the pROC package to evaluate the diagnostic capability of CD161. The significance of the correlation between the two groups was assessed by Pearson correlation analysis. The prognostic value was evaluated using Kaplan Meier curves, and the significance of the prognostic value was tested by a log-rank test. A P-value of < 0.05 was deemed statistically significant.

Results

Patient characteristics

A total of 109 patients who underwent liver resection for HCC were included in this study, with their details summarized in Table 1. The median age at diagnosis was 59 years (range: 32–86 years). Most patients were male (81.7%), had a history of Hepatitis B/C (90.8%), had a solitary tumor (81.7%), and possessed a tumor size of ≤ 5 cm (60.6%). The majority of individuals in this patient cohort were classified as TNM stages I–II (80.7%) and had an ALBI grade of 1 (76.1%). As of the cut-off date of May 1, 2024, two patients were lost to follow-up, and three patients died from other causes. Ultimately, a total of 104 patients completed the follow-up with a median follow-up duration of 35.4 months.

Table 1 Correlation between CD161 expression and clinicopathologic features in HCC patients.

Variables	Stratification	Frequency	CD161 High (n = 62)	CD161 low (n = 47)	P value	
Age	≤60	63 (57.8%)	37	26	0.648	
	>60	46 (42.2%)	25	21		
Gender	Male	89 (81.7%)	52	37	0.492	
	Female	20 (18.3%)	10	10		
Hepatitis B/C	Positive	99 (90.8%)	55	44	0.379	
	Negative	10 (9.2%)	7	3		
Tumor size	≤5 cm	66 (60.6%)	43	23	0.031	
	>5 cm	43 (39.4%)	19	24		
Tumor numbers	Solitary	89 (81.7%)	51	38	0.851	
	Multiple	20 (18.3%)	11	9		
ALBI grade	1	83 (76.1%)	47	36	0.924	
	2	26 (23.9%)	15	11		
AFP	≤400 ng/ml	78 (71.6%)	45	33	0.786	
	>400 ng/ml	31 (28.4%)	17	14		
NLR	≤3	58 (53.2%)	34	24	0.696	
	>3	51 (46.8%)	28	23		
PLR	≤130	55 (50.5%)	34	21	0.294	
	>130	54 (49.5%)	28	26		
TNM stage	I/II	88 (80.7%)	54	34	0.053	
	III	21 (19.3%)	8	13		
CNLC stage	I/II	95 (87.2%)	57	38	0.087	
	III	14 (12.8%)	5	9		
BCLC stage	A	83 (76.2%)	52	31	0.090	
	B	12 (11.0%)	5	7		
	C	14 (12.8%)	5	9		
PVTT	Absent	96 (88.1%)	58	38	0.043	
	Present	13 (11.9%)	4	9		
MVI	Absent	96 (88.1%)	56	40	0.405	
	Present	13 (11.9%)	6	7		

Expression of CD161 is lower in HCC tissues

The mRNA levels of CD161 were assessed in paired tumor and normal liver tissues across TCGA, ICGC, and GSE14520 datasets. Low CD161 expression was observed in the tumor compared to normal liver tissue in the three databases (Figs. 1A–1C). To validate these findings at the protein level, a total of 109 liver cancer tissues from 109 HCC patients, along with matched adjacent normal tissues from 53 of these patients, were collected for the assessment of CD161 protein expression through immunohistochemistry. IHC analysis of HCC tissues showed that CD161 was mainly expressed in the cytoplasm and cell membrane, consistent with a previous study (Xu et al., 2023). Among the 109 liver cancer tissues in this cohort, 103 (94.5%) exhibited positive expression of CD161. Additionally, positive CD161 expression was observed in all 53 adjacent normal tissues. Representative IHC staining images displayed different levels of CD161 expression in both tumor and adjacent normal tissues (Figs. 2A–2J). The Human Protein Atlas database (http://www.proteinatlas.org/pathology) was also used to investigate protein levels of CD161 in HCC (Fig. 2K). The immunohistochemical results showed that CD161 was significantly upregulated in 53 adjacent non-cancerous tissues compared to 109 liver cancer tissues (Fig. 1E). Additionally, in the 53 matched pairs of HCC and adjacent normal tissues, CD161 was downregulated in cancerous tissues compared to adjacent normal tissues (Fig. 1D). Furthermore, the diagnostic capability of CD161 was evaluated by analyzing the ROC curve. The area under the curve (AUC) values were 0.658, 0.689, 0.792, and 0.808 for TCGA, ICGC, GSE14520, and the study cohort, respectively, suggesting a robust diagnostic potential for CD161 in HCC (Figs. 1F–1I).

Figure 1 The difference in CD161 expression between normal liver tissue and HCC tissue.

(A–C) The mRNA expressions of CD161 in paired normal liver tissue and HCC tissue in the TCGA (n = 50), ICGC (n = 193) and GSE14520 databases (n = 213). (D) The protein expressions of CD161 in paired normal liver tissue and HCC in the study cohort (n = 53). (E) The protein expressions of CD161 in normal liver tissue (n = 53) and all liver cancer tissue (n = 109) in the study cohort. (F–I) ROC curves for CD161 expression in normal liver tissue and HCC in TCGA, ICGC, GSE14520, and the study cohort. **P < 0.01 and ****P < 0.0001.

Figure 2 Representative immunohistochemistry staining for CD161.

(A, B) Negative staining in tumor tissue at magnifications x100 and x400. (C, D) Weak staining in tumor tissue at magnifications x100 and x400. (E, F) Moderate staining in tumor tissue at magnifications x100 and x400. (G, H) Strong positive staining in tumor tissue at magnifications x100 and x400. (I, J) CD161 exhibited strong staining in adjacent normal tissues at magnifications x100 and x400. (K) Protein expressions of CD161 in human normal liver tissue and HCC (Human Protein Atlas database).

Expression of CD161 is associated with tumor size and PVTT

The associations between CD161 expression and clinicopathological characteristics were also investigated. Using a median immunohistochemical score of 5.6 as the threshold, 62 patients (56.9%) were categorized as exhibiting high CD161 expression, while the remaining 47 cases (43.1%) were classified as having low CD161 expression. As shown in Table 1, CD161 expression was positively correlated with smaller tumor sizes (≤5 cm) and the absence of portal vein tumor thrombus. The relationship between CD161 expression and clinical pathological features was further confirmed in the TCGA and GSE14520 datasets (detailed information can be found in Table S1). Higher levels of CD161 expression were typically observed in patients with smaller tumors and was linked to absence of PVTT in HCC, as indicated by the results from analyzing multiple public and in-house cohorts. Furthermore, higher levels of CD161 expression were also observed in surviving patients in comparison to deceased patients (Fig. 3).

Figure 3 The association between CD161 expression and clinicopathological characteristics of HCC patients.

(A) CD161 was significantly increased in HCC patients with tumor size ≤ 5 cm in the study cohort. (B) CD161 was significantly increased in HCC patients without PVTT in the study cohort. (C–E) CD161 was highly expressed in HCC patients with earlier TNM (C), CNLC (D), and BCLC (E) stage in the study cohort. (F) CD161 was significantly increased in surviving HCC patients in the study cohort. (G) CD161 was significantly increased in HCC patients with earlier TNM in TCGA. (H) CD161 was significantly increased in surviving HCC patients in TCGA. (I, J) Increased CD161 levels in HCC patients were non-significantly associated with earlier TNM and BCLC stage in GSE14520. (K) CD161 was significantly increased in HCC patients with tumor size ≤ 5 cm in GSE14520. (L) CD161 was significantly increased in surviving HCC patients in GSE14520.

CD161 is closely linked to a favorable prognosis in HCC patients

CD161’s prognostic significance was investigated in our cohort as well as in the TCGA and GSE14520 databases. In the TCGA and GSE14520 databases, the median value of CD161 expression was selected as the cutoff value to categorize HCC patients into CD161 high-expression and low-expression groups within each database. Based on a Kaplan–Meier analysis, the CD161 high-expression groups had longer OS and RFS compared to the low-expression groups (Fig. 4), although RFS analysis in the GSE14520 dataset did not reach statistical significance. Overall, these findings suggested that CD161 conferred a protective effect on HCC patients. To further elucidate the role of CD161 in the prognosis of HCC patients, univariable and multivariable Cox analyses were performed. Univariate analysis of the Cox proportional hazards model showed that the factors significantly associated with OS and RFS were tumor size, tumor number, PVTT, TNM stage, CNLC stage, AFP level, and CD161 expression. The clinicopathological parameters that were screened out in the univariate analysis (p < 0.05) were then included in the multivariate survival analysis. The results indicated that TNM stage, AFP level, and CD161 expression were independent prognostic factors for predicting RFS and OS (Table 2).

Figure 4 Kaplan–Meier analysis of CD161 expression in the study cohort and in the TCGA and GSE14520 databases.

(A–C) CD161 expression was positively correlated with OS in the study cohort (A), TCGA database (B), and GSE14520 database (C). (D–F) CD161 expression was positively correlated with RFS in the study cohort (D), TCGA database (E), and GSE14520 database (F).

Table 2 Univariate and multivariate analysis of risk factors for overall survival and recurrence free survival.

Variable	Overall survival	Relapse free survival	
	Univariate analysis	Multivariate analysis	Univariate analysis	Multivariate analysis	
	HR	95% CI	P	HR	95% CI	P	HR	95% CI	P	HR	95% CI	P	
Age (>60 vs≤60)	0.854	0.373–1.955	0.710				1.211	0.702–2.089	0.491				
Gender (Male vs Female)	0.775	0.287–2.092	0.615				0.823	0.423-1.604	0.568				
Hepatitis (B/C) (Yes vs No)	0.654	0.194–2.202	0.493				1.225	0.441–3.403	0.696				
Tumor size (cm) (>5 vs≤5)	3.148	1.356–7.306	0.008				2.174	1.259–3.753	0.005				
Tumor number (Multiply vs Solitary)	3.491	1.503–8.109	0.004				2.369	1.259–4.457	0.007				
PVTT (Present vs Absent)	10.308	4.208–25.253	0.000				4.275	2.098–8.709	0.000				
MVI (Present vs Absent)	1.850	0.618–5.538	0.271				1.223	0.518–2.886	0.646				
TNM stage (AJCC) (I/II vs III)	14.687	5.779–37.329	0.000	7.887	2.990–20.802	0.000	5.424	2.905–10.127	0.000	4.267	2.232–8.157	0.000	
CNLC stage (I/II vs III)	11.555	4.724–28.267	0.000				4.267	2.142–8.499	0.000				
BCLC stage													
A													
B	3.206	0.955–10.762	0.059				3.183	1.508–6.720	0.002				
C	13.975	5.435–35.935	0.000				5.061	2.492–10.278	0.000				
NLR (>3 vs≤3)	0.562	0.238–1.327	0.189				0.763	0.439–1.326	0.337				
PLR (>130 vs≤130)	1.178	0.515–2.691	0.698				0.843	0.488–1.456	0.541				
ALBI (Grade 2 vs 1)	1.888	0.800–4.456	0.147				2.142	1.176–3.900	0.013				
AFP (ng/ml) (>400 vs≤400)	6.637	2.796–15.756	0.000	4.444	1.827–10.809	0.001	3.189	1.802–5.644	0.000	3.059	1.715-5.455	0.000	
CD161 (High vs Low)	0.256	0.105–0.625	0.003	0.397	0.159–0.991	0.048	0.440	0.253–0.764	0.004	0.527	0.299–0.926	0.026	

CD161 is involved in the immune response and immune regulation in HCC

A Pearson correlation analysis (|R| > 0.5, P < 0.05) was used to identify the genes most related to CD161 in both the TCGA and ICGC databases. GO and KEGG analyses were conducted based on these gene sets. In the TCGA database, the biological processes most closely linked to CD161 included immune response, adaptive immune response, T cell activation, and the T cell receptor signaling pathway (Fig. 5A). Additionally, the cellular components most associated with CD161 were the plasma membrane and external side of the plasma membrane (Fig. 5B). The molecular function most associated with CD161 was transmembrane signaling receptor activity (Fig. 5C). The signaling pathways most closely associated with CD161 included cytokine-cytokine receptor interaction and the chemokine signaling pathway (Fig. 5D). The biological functions, cellular components, molecular processes, and signaling pathways most associated with CD161 in the ICGC database mirrored those in the TCGA database (Figs. 5E–5H).

Based on the results of the GO and KEGG analyses, CD161 was considered to be crucial for immune response, T cell activation, cytokine-cytokine receptor interaction, and chemokine signaling. Therefore, an in-depth investigation into the effects of CD161 on these cellular functions and signaling pathways was conducted. Gene set variation analysis (GSVA) was used to determine the enrichment score of the immune process in both the TCGA and ICGC databases. The correlation analysis revealed a positive association between CD161 expression and the majority of immune responses, as well as between CD161 expression and functions and signaling pathways related to cytokines and chemokines (Fig. 6A). Subsequent validation in the ICGC database confirmed these results (Fig. 6B). These findings suggested that CD161 likely plays a pivotal role in immune response and immune regulation in HCC.

Figure 5 The functional enrichment analysis of CD161 in HCC.

(A–D) The biological processes (BP), cellular components (CC), molecular functions, and Kyoto Encyclopedia of Genes and Genomes (KEGG) pathway most associated with CD161 in the TCGA database. (E–H) The biological processes (BP), cellular components (CC), molecular functions, and Kyoto Encyclopedia of Genes and Genomes (KEGG) pathway most associated with CD161 in the ICGC database.

Figure 6 The correlation analysis between CD161 expression and immune function enrichment scores.

(A) The expression of CD161 was positively correlated with immune function enrichment scores in the TCGA database; (B) The expression of CD161 was positively correlated with immune function enrichment scores in the ICGC database.

CD161 is related to an inflamed tumor microenvironment in HCC

Considering that CD161 was associated with multiple immune-related processes, the specific immunological role of CD161 in HCC was investigated. The median value of CD161 expression from the TCGA and ICGC databases was selected as the threshold to categorize CD161 expression into high and low groups. A majority of chemokines and receptors (Fig. 7A), cytokines and receptors (Fig. 7B), major histocompatibility complex (MHC) molecules (Fig. 8A), and co-stimulators (Fig. 8B) were notably correlated with CD161 in HCC. To explore the association between CD161 and immunophenotyping, CD161 expression was compared across different immune subtypes based on previously published research (Thorsson et al., 2018). The findings indicated that CD161 expression peaked in the IFN-γ dominant subtype, followed by the inflammatory subtype, and was lowest in the lymphocyte depleted subtype (Fig. 9A). Subsequently, comparisons were made among different CD161 groups regarding the leukocyte fraction, lymphocyte infiltration signature score, BCR Shannon score, and TCR Shannon score. Higher values were observed for all of these metrics in the group with high CD161 expression (Figs. 9B–9D). Taken together, CD161 is positively associated with an inflamed tumor microenvironment in HCC.

Figure 7 The relationship between CD161 expression and chemokines and cytokines along with their receptors.

(A) CD161 expression was positively correlated with most chemokines and their receptors. (B) CD161 expression was positively correlated with most cytokines and their receptors. *P < 0.05, **P < 0.01, ***P < 0.001, and ****P < 0.0001.

Figure 8 The correlations between CD161 expression and MHCs and co-stimulators.

(A) The expression of CD161 was positively associated with MHCs. (B) The expression of CD161 was positively associated with co-stimulators. *P < 0.05, **P < 0.01, ***P < 0.001, and ****P < 0.0001.

Figure 9 The correlations between CD161 expression and immunophenotyping and immunological markers in HCC.

(A) The expression of CD161 in different immune subtypes. (B–D) The leukocyte fraction (B), lymphocyte infiltration signature score (C), BCR Shannon score (D), and TCR Shannon score (D) were elevated in the CD161-high expression group. **P < 0.01 and ****P < 0.0001.

CD161 is associated with high immune cell infiltration

To explore the relationship between CD161 expression and immune cell infiltration, a correlation analysis was conducted using xCell, CIBERSORT, and the MCP-counter method in both the TCGA and ICGC databases. A positive association between CD161 expression and the infiltration levels of most immune cells, especially DC cells, CD8+ T cells, and CD4+ T cells, was revealed by xCell in the TCGA database (Fig. 10A). The xCell analysis also found that the CD161 high-expression group exhibited significantly higher immune scores, stroma score, and microenvironment score (Figs. 10B–10C) in the TCGA database. The X cell analysis in the ICGC database produced consistent results with those from the TCGA database (Fig. 11). Similarly, CIBERSORT (Figs. 12A–12B) and MCP-counter (Figs. 12C–12D) analyses also revealed a positive correlation between CD161 expression and the infiltration levels of DC cells, CD8+ T cells, and CD4+ T cells. These findings suggest that CD161 is related to high immune cell infiltration in HCC. This further indicates the positive correlation between CD161 and the inflammatory immune microenvironment, suggesting its potential as a biomarker for immunotherapy.

Figure 10 The immune cell infiltration analysis of CD161 based on xCell in the TCGA database.

(A) CD161 had a positive correlation with the levels of DC cell, CD8+T cell, and CD4+T cell infiltration in the TCGA database (xCell analysis). (B–D) The CD161 high-expression group exhibited significantly higher immune scores, stroma score, and microenvironment score compared to the low-expression group in the TCGA database (xCell analysis). *P < 0.05, **P < 0.01, ***P < 0.001, and ****P < 0.0001.

Figure 11 The immune cell infiltration analysis of CD161 based on xCell analysis in the ICGC database.

(A) CD161 had a positive correlation with the levels of DC cell, CD8+T cell, and CD4+T cell infiltration in the ICGC database (xCell analysis). (B–D) The CD161 high-expression group exhibited significantly higher immune scores, stroma score, and microenvironment score compared to the low-expression group in the ICGC database (xCell analysis). *P < 0.05, **P < 0.01, ***P < 0.001, and ****P < 0.0001.

Figure 12 The immune cell infiltration analysis of CD161 based on CIBERSORT and MCP counter analysis.

(A, B) CD161 had a positive correlation with the levels of DC cell, CD8+T cell, and CD4+T cell infiltration based on CIBERSORT analysis in the TCGA and ICGC databases. (C, D) CD161 had a positive correlation with the levels of T cell, CD8+T cell, cytotoxic lymphocytes, and DC cell infiltration based on MCP counter analysis in the TCGA and ICGC databases. *P < 0.05, **P < 0.01, ***P < 0.001, and ****P < 0.0001.

CD161 predicts immunotherapeutic responses

The above findings demonstrated that CD161 expression was positively associated with an inflamed TIME and immune cell infiltration, which raises the question of its possible use as a biomarker to predict the efficacy of immunotherapy. The relationship between CD161 expression and previously published immune-related signatures (Spranger, Bao & Gajewski, 2015; Danilova et al., 2016; Ayers et al., 2017; Hsu et al., 2021), potentially aiding in predicting responses to immunotherapy across different cancers, was analyzed using GSVA. The findings revealed a positive correlation between the expression of CD161 and these predictive genetic signatures (Figs. 13A–13B). Furthermore, a robust correlation was observed between CD161 and prominent inhibitory immune checkpoints including PD-1, PD-L1, TIM3, LAG3, CTLA-4, TIGIT, and CD200R1 across the TCGA and ICGC databases (Figs. 14A–14B). Subsequent exploration was conducted to examine the relationship between CD161 expression and treatment response in an immunotherapy cohort (GSE140901) comprised of 24 HCC patients. The CD161 high-expression group had a higher proportion of patients evaluated as partial response (PR) and stable disease (SD), while the low-expression group had a higher proportion of patients assessed as progressive disease (PD) (Fig. 14C). However, there was no significant statistical difference in the objective response rates (ORR) and disease control rates (DCR) in the GSE140901 set (Table S2). In addition, the association between CD161 expression and the efficacy of immunotherapy was investigated in the IMvigor 210 cohort. CD161 expression peaked in the inflamed subtype and was lowest in the desert subtype within the IMvigor 210 cohort (Fig. 14D). Furthermore, CD161 expression exhibited a positive correlation with the level of IHC-assessed PD-L1 staining on immune cells (IC; Fig. 14E). In the IMvigor 210 cohort, the CD161 high-expression group had a larger percentage of patients with complete response (CR), partial response, and stable disease, and the CD161 low-expression group had a higher percentage of patients experiencing progressive disease (Fig. 14F). Additionally, the CD161 high-expression group exhibited superior objective response rates and disease control rates compared to the CD161 low-expression group, despite the lack of statistical significance in ORR (Table 3).

Figure 13 The expression of CD161 was linked to gene signatures associated with immunotherapy response.

(A, B) The expression of CD161 had a positive correlation with immune-related signatures, helping predict the response to immunotherapy in various cancers in the TCGA (A) and ICGC (B) databases. *P < 0.05, **P < 0.01, ***P < 0.001, and ****P < 0.0001.

Figure 14 The value of CD161 in predicting immunotherapeutic responses.

(A, B) CD161 expression was positively associated with common immune checkpoints in the TCGA and ICGC databases. (C) The distribution of patients assessed as PR, SD, and PD among different CD161-expression groups in GSE140901. (D) The expression of CD161 was highest in the inflamed subtype and lowest in the desert subtype based on the IMvigor 210 cohort. (E) The expression of CD161 had a positive association with the level of IHC-assessed PD-L1 staining on immune cells (IC) in the IMvigor 210 cohort. (F) The distribution of patients assessed as CR, PR, SD, and PD among different CD161-expression groups in the IMvigor 210 cohort. *P < 0.05, **P < 0.01, ***P < 0.001, and ****P < 0.0001.

Discussion

CD161, a C-type lectin-like receptor, is primarily found in the bone marrow, spleen, and other tissues, exhibiting elevated levels in mammary lymph nodes and tumors (Fahmi et al., 2010). Previous studies have highlighted a strong link between CD161 and infectious diseases including tuberculosis, hepatitis B, and human papillomavirus (HPV; Duurland et al., 2022; Liu et al., 2023; Jiang et al., 2023). Recently, there has been growing interest in the role CD161 plays in tumors, especially in modulating the tumor immune microenvironment. Several pan-cancer studies have shown CD161 to be a potential prognostic and immunological marker in multiple tumor types (Ye et al., 2021; Zhou et al., 2021; Li et al., 2022). Nevertheless, the biological function and clinical significance of CD161 in HCC have not been completely elucidated. This study details a comprehensive analysis, for the first time, of CD161 expression and its related biological functions, prognostic implications, and predictive role in immunotherapy efficacy in HCC. The analyses showed that CD161 was highly expressed in adjacent normal tissues and associated with smaller tumor size and absence of portal vein tumor thrombus. CD161 was also related to a favorable prognosis in HCC in terms of OS and RFS. Moreover, an enrichment analysis found that CD161 was significantly correlated with the immune response as well as cytokine and chemokine signaling pathways, suggesting CD161’s involvement in immune regulation during cancer progression. Specifically, CD161 showed a positive association with immunomodulators and tumor-infiltrating immune cells. Finally, multiple public databases showed that patients with high expression of CD161 were more likely to benefit from immunotherapy.

This study revealed that CD161 expression was elevated in normal liver tissue compared to tumor tissue at both the mRNA and protein level. Previous study has also shown that KLRB1 mRNA is significantly downregulated in most cancer tissues compared to their corresponding normal tissues (Cheng et al., 2022). Our study utilized immunohistochemistry to demonstrate low expression of CD161 in liver cancer tissues at the protein level. This study also demonstrated the significant value of using CD161 as a biomarker in diagnosing HCC through its patient cohort as well as multiple public databases. In line with the current study, Xu et al. (2023) highlighted the strong diagnostic potential of CD161 in breast cancer. Additionally, CD161 was notably enriched in HCC patients with smaller tumor sizes (≤5 cm) and the absence of PVTT. Moreover, patients in the high-expression CD161 group exhibited extended OS and RFS, with CD161 being an independent protective factor affecting HCC prognosis. Gentles et al. (2015) identified KLRB1 (CD161) as a favorable prognostic gene through an investigation of expression profiles across 18,000 human tumors encompassing 39 malignancies. Another study (Pan et al., 2020) demonstrated that KLRB1 was a potential pivotal gene linked to a more favorable prognosis in HCC. The findings of the present study indicated a close correlation between low CD161 expression and adverse clinicopathological characteristics, along with a poor prognosis of HCC in multiple public and in-house cohorts.

Table 3 Therapeutic efficacy of response in IMvigor 210.

	High group (n = 149)	Low group(n = 149)	P value	
CR	14	11		
PR	22	21		
SD	39	24		
PD	74	93		
ORR	24.16%	21.48%	0.581	
DCR	50.33%	37.58%	0.027	

The biological function analysis revealed that CD161 was significantly linked to immune response as well as cytokine and chemokine signaling pathways in HCC. Previous research has indicated that CD4+CD161+ and CD8+CD161+ T cells are capable of releasing a higher amount of proinflammatory cytokines compared to CD161-negative cells (Cosmi et al., 2008; Santegoets et al., 2019). In addition, another study demonstrated that CD161 labels a subset of cytotoxic T lymphocytes (CTLs) that maintain strong immune reactivity (Wei et al., 2023). Furthermore, single-cell data demonstrated elevated KLRB1 expression in tissue-resident NK and T cells within HCC, which co-expressed markers of immune activation (Fang & Zhou, 2024). These findings further confirmed and cross-validated the conclusions from the present study’s GSVA. Notably, CD161 is emerging as a novel immune checkpoint in tumorigenesis. Its ligand, lectin-like transcript 1 (LLT1), expressed in various tumors, interacts with KLRB1 on immune cells, thereby modulating the immune milieu within tumor microenvironments. To delve deeper into the relationship between CD161 and the TIME, multiple databases were used to analyze immune infiltrating cells. A positive association of CD161 expression with the number of infiltrating CD8+ cells, CD4+ cells, and DCs was observed. These findings indicate that CD161 is closely correlated with immune infiltration and plays a significant role in regulating the tumor immune microenvironment in HCC.

This study found that high expression of CD161 indicated a favorable prognosis in HCC. However, the precise molecular mechanism of CD161 in the development of HCC remains unclear. Li et al. (2020) constructed a single-cell immune landscape from 15 pairs of HCC tumors, revealing that CD8+PD1+CD161+ T cells exhibited heightened immune activity, which may exert an increased anti-tumor effect and yield a more favorable prognosis. This finding aligns with a recent study (Fang & Zhou, 2024) that suggested a more favorable prognosis in HCC patients with elevated expression levels of KLRB1 in the CD8+ T cells and NK cells. The present study’s GSVA analysis from the TCGA and ICGC databases revealed a positive correlation between CD161 and T cell activation, immune responses, and cytokine and chemokine signaling pathways, implying a role in promoting immune activation. This evidence suggests that CD161 may exert inhibitory effects on the development of HCC by enhancing anti-tumor immune responses. Conversely, Sun et al. (2021) discovered that CD8+ T cells in recurrent tumors overexpressed KLRB1 (CD161) compared to those in tumors of primary HCC patients. Their study suggested that CD161+CD8+ T cells exhibited a dysfunctional cytotoxicity and low expansion phenotype, which may compromise antitumor immunity and lead to a poor prognosis. This difference may be closely related to the heterogeneity of liver cancer, the complexity of the tumor microenvironment, and the dynamic regulation of tumor immunity. Further in-depth research is necessary to elucidate the specific mechanisms involved.

CD161 exhibits varying roles in different tumors. Breast invasive carcinoma (BRCA) patients with low KLRB1 levels were linked to advanced disease stages, poor prognosis, and a decreased survival probability compared to those with high KLRB1 expression. Further mechanistic studies have shown that CD161 can inhibit the proliferation, invasion, and migration of breast cancer cells. Decreased KLRB1 expression in BRCA may compromise cancer immunity and lead to an unfavorable prognosis (Huang et al., 2023; He et al., 2023). Similarly, CD161 interacts with its receptor LLT1 to regulate the oral squamous cell carcinoma (OSCC) tumor immune microenvironment, thereby improving the prognosis of OSCC patients (Hu et al., 2024). Conversely, CD161 plays a divergent role in other tumor types. Researchers found that CD161 can suppress T cell anti-tumor immunity in glioma, and knocking down KLRB1 or using an antibody-mediated blockade of CD161 can enhance the ability of T cells to kill tumors (Mathewson et al., 2021). Likewise, another study showed that targeting the inhibitory CD161 receptor could bolster T cell-mediated immunity against hematological malignancies (Alvarez Calderon et al., 2024). The reasons for these differences are complex. On one hand, the interaction of CD161/LLT1 on T cell surfaces displays remarkable diversity, delivering both co-stimulatory and co-inhibitory signals linked to various immunopathological microenvironments (Germain et al., 2011); Wyrożemski & Qiao, 2021). Due to variations in the immune microenvironments in different tumors, this disparity influences the interaction mode of CD161/LLT1, resulting in an opposite effect. On the other hand, there are significant differences in cancer cell type, tissue samples, and methodology, which can also lead to different results. The summary of CD161’s role in various types of cancers, as well as its potential molecular mechanisms, is presented in Table S3.

Immune checkpoint inhibitors can activate immune cells and enhance their antitumor abilities by disrupting tumor-associated pathways that inhibit immune cell activation. Several immune checkpoint inhibitors have been approved by the FDA as a monotherapy or in combination for the treatment of advanced HCC based on trial efficacy data. Due to the complexity of the tumor immune microenvironment of HCC, the efficacy of immunotherapy in HCC is limited. The current study showed that CD161 expression correlated with high levels of immune cell infiltration and was notably enriched in the inflammatory immune subtype, characteristics that indicate an immune-hot TIME and suggest its responsiveness to immunotherapy. Conversely, CD161 showed the lowest expression in the lymphocyte depleted subtype, suggesting an immune-cold TIME. Additionally, a significantly higher number of patients assessed with progressive disease were observed in the CD161 high-expression group compared to those in the CD161 low-expression group in the two immunotherapy cohorts. Consistent with this research, Tang et al. (2022) identified CD161 as a tumor immunological phenotype-related gene capable of predicting prognosis as well as the effectiveness of immunotherapy in HCC. Mechanistically, a positive correlation was revealed in this study between CD161 expression and the quantity of CD8+ T cells, which have been demonstrated as promising biomarkers for immunotherapy in various malignant tumors (He et al., 2021). There was no significant statistical difference in the ORR and DCR in the GSE140901 set, which could be attributed to the limited number of patients included. Overall, it is reasonable to assume that CD161 likely plays a predictive role in immunotherapy, and further research should be undertaken to substantiate this hypothesis.

Despite providing new insights into the role of CD161 in HCC, the current study still has some limitations. Firstly, the sample size of enrolled participants in this study is relatively small, necessitating multicenter large-sample research. Second, the data in this study were primarily collected from clinical cases; in vivo and in vitro experiments have not been carried out to elucidate the role of CD161 in HCC. Third, most HCC patients are diagnosed at advanced stages, thereby missing the opportunity for curative surgery. This limitation does, to some extent, restrict the utility of CD161 in assessing the prognosis of patients undergoing liver cancer surgery. Nevertheless, liver biopsy samples can serve as an alternative for detecting CD161 expression, rendering this biomarker useful for prognostic evaluation in patients with advanced HCC. Finally, the validation of CD161’s role in HCC immunotherapy has been limited to small sample public databases. Our next step will involve conducting large-scale clinical studies to further validate these findings.

Conclusions

In conclusion, this study conducted a comprehensive assessment of CD161, indicating its correlation with a favorable prognosis and immune infiltration in hepatocellular carcinoma, exploring its regulatory role in the immune microenvironment, and highlighting its potential as a prospective biomarker for immunotherapy. Further in vivo and in vitro studies are required to elucidate the biological functions and specific mechanisms of CD161 in HCC.

Supplemental Information

Supplemental Information 1 Summary of clinicopathologic data in TCGA and GSE14520

Supplemental Information 2 Therapeutic efficacy of the different group in GSE140901

Supplemental Information 3 The summary of CD161’s role in different types of cancers and its potential molecular

Supplemental Information 4 Abbreviations

We acknowledge the TCGA, ICGC, and GEO databases for providing their platforms, and we thank contributors for uploading their datasets. In addition, we acknowledge the invaluable help of peer reviewers.

Additional Information and Declarations

Competing Interests

Author Contributions

Human Ethics

Data Availability

The authors declare there are no competing interests.

Jinfeng Wang conceived and designed the experiments, performed the experiments, analyzed the data, prepared figures and/or tables, authored or reviewed drafts of the article, and approved the final draft.

Xiaoxiao Wang performed the experiments, prepared figures and/or tables, and approved the final draft.

Jianfei Shi conceived and designed the experiments, analyzed the data, authored or reviewed drafts of the article, and approved the final draft.

Yongjun Wang performed the experiments, prepared figures and/or tables, and approved the final draft.

Lili Mi performed the experiments, analyzed the data, authored or reviewed drafts of the article, and approved the final draft.

Man Zhao analyzed the data, prepared figures and/or tables, and approved the final draft.

Guangjie Han analyzed the data, prepared figures and/or tables, and approved the final draft.

Fei Yin conceived and designed the experiments, authored or reviewed drafts of the article, and approved the final draft.

The following information was supplied relating to ethical approvals (i.e., approving body and any reference numbers):

The Fourth Hospital of Hebei Medical University granted Ethical approval to carry out the study within its facilities (certificate no. 2024KS003).

The following information was supplied regarding data availability:

The raw data and code are available at Zenodo: sunnydream1989. (2024). sunnydream1989/CD161: v1 (Version v1). Zenodo. https://doi.org/10.5281/zenodo.14249354.

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
