# Peer review of "CD161, a promising prognostic biomarker in hepatocellular carcinoma, correlates with immune infiltration"

_PeerJ, doi:10.7717/peerj.19055_

## Round 0.1 · original submission · Major Revisions

Please address issues pointed by all reviewers and amend manuscript accordingly.

Reviewer 1 ·

Basic reporting

This study explores CD161 as a potential prognostic biomarker for hepatocellular carcinoma (HCC), assessing its expression in tumor and adjacent normal tissues and examining its correlation with clinicopathological factors, immune cell infiltration, and survival outcomes. The study involves immunohistochemistry and bioinformatic analyses across multiple datasets, revealing that high CD161 expression is associated with smaller tumor size, earlier stages, and favorable outcomes (extended overall survival and recurrence-free survival). CD161 is closely related to immune cell infiltration, particularly with T cells and dendritic cells, suggesting its role in regulating the tumor immune microenvironment. Notably, high CD161 expression aligns with better responses to immunotherapy, implying its potential predictive value for therapeutic efficacy.

Experimental design

1.The study's cohort is limited to 109 HCC patients, which restricts the robustness of its conclusions.
2. Findings are primarily based on clinical data; no in vivo or in vitro experiments were performed to investigate CD161's biological functions in HCC.
3.In immunohistochemical analysis data, tissues with a high proportion of positive cells will also have higher staining intensity. Does using the product of the two as the immune response score affect the critical value?

Validity of the findings

1. There was no statistically significant correlation between TNM staging and CD161 expression between the queue you created and the queue in GSE14520 database. How did you determine that TNM staging is positively correlated with CD161?
2. And there is an error in the grouping of CD161 expression in your queue. What is the basis for determining that RFS is positively correlated with CD161?
3.Among the 103 positive cases of CD161 protein expression, how many were in HCC tissue and how many were in adjacent normal tissue? Why was no comparison made?

·

Basic reporting

1- The article was written in unambiguous, professional English used throughout.
2- The article included sufficient background and the references are suitable and updated.
3- The tables are well-structured.
4- The figures are well-structured but the quality needs to be improved.
5- To Figure 1, add the p-value on the related ROC curves. Additionally, you should explain the difference between D and E figures.

Experimental design

1- The original primary research within the Aims and Scope of the journal.
2- The research question is well-defined, relevant & meaningful.
3- The investigations performed are rigorous to a high technical & ethical standard.
4- The methods are described with sufficient detail & information to replicate.
5- An important point that needs to be highlighted is the management introduced for the selected patients. As the authors described in the 1st table, 14 patients were BCLC-C, and 13 cases presented with PVTT. According to the previously mentioned data, these cases are not candidates for curative surgical resection.
6- At line 102, add a suitable reference for the CNLC and BCLC staging.
7- Kindly varify this sentence "To validate these findings 196 at the protein level, a total of 109 HCC tissues, comprising 53 HCC and their respective adjacent 197 normal tissues, were collected for the assessment of CD161 protein expression through 198 immunohistochemistry methods." to be clear.

Validity of the findings

1- The results are novel and have a good impact on the field of HCC management.
2- The underlying data are robust and statistically sound.
3- The conclusions are well-stated, linked to the original research question, and limited to supporting results.

Additional comments

An important point that should be pointed out is the limited use of the studied biomarker (Expression of CD161 in HCC tissues) in the assessment of HCC patients who underwent surgical resection which is not usually the suitable treatment plan in HCC patients. It is well known that most HCC patients are discovered lately. In addition, it could be used on HCC tissue from liver biopsy.

Reviewer 3 ·

Basic reporting

Overall I found the article is easy to follow and clear in conclusion. The main and supplementary data are overall comprehensive and sufficient for understanding the work. However just a few things to enhance:
1. Conclusion: The author wrote “CD161 emerges as a promising prognostic biomarker for HCC.” Would suggest they also include the direction (CD161 expression seems to imply positive prognosis, different from some of the previous work).
2. Discussion: I would encourage the authors to include more discussions over the potential molecular mechanism for CD161, beyond just clinical data correlation.
3. Table S2: There is a missing word in the title.

Experimental design

The authors have focused on CD161 expression in HCC and potential mechanism behind, as studies in other cancer types shed light on CD161. The approach of this work is a combination of clinical sample analysis together with computational analysis of different public datasets. I found the problem to solve and analysis per se are largely clear; some points to refine as below:
1. Line 72-77, 345-348: The authors mentioned there are papers with conclusion in a different direction. Would be great to see in the later discussion section that, why there are different conclusions (e.g. cell type specificity, or due to different methodology used). Even ore helpful is to include a table in supplementary comparing different studies (e.g. which direction; what cancer type; which tissue examples; methodology used).
2. Line 126-128: What is the exact tissue profiled in the public dataset? Core tumor tissue or also include adjacent tissue? Better to have some clarifications for all external datasets used in this study if possible (given it is known that expression level will differ between tumor and adjacent). Similarly in the later writing throughout the article, clarify as much as possible.
3. Line 162-164: The authors mentioned “The levels of immune cell infiltration were compared between different groups, which were stratified based on the median CD161 expression level in the TCGA and ICGC databases.” Would like to know how this stratification is conducted.
4. Line 212-214: Why does the author picked the threshold of 5.6? In description it seems to be just median score but curious whether there is a standard scoring for this.
5. Line 220: Maybe should not include TNM here (given the next sentence mentioned TNM is not statistically significant).
6. Line 252 to 261: Would benefit a bit if there are more molecular mechanism literatures cited at the end to cross-validate the conclusions from GSVA.
7. On top of the databases used, there is a paper on Cell in 2021 titled “Single-cell landscape of the ecosystem in early relapse hepatocellular carcinoma”. This paper could be relevant (for example KLRB1/ CD161 is involved) but not cited in the current work. Have the authors looked into the conclusions of that work? Not necessarily re-analyzing that data, but more to see if anything relevant.

Validity of the findings

The findings from clinical sample analysis and computational analysis are clear. The authors also highlight the caveat at the end that “the data in this study primarily stem from clinical cases; in vivo and in vitro experiments have not been carried out to elucidate the role of CD161 in HCC.” Understood that it may be difficult to request experimental validation if capability not allowed, but I will encourage the authors to:
1. Think about potential molecular mechanism (not just correlation) of CD161 in HCC based on scanning existing literature (there are some papers like the 2021 Cell paper I mentioned above that could provide some references, beyond just GSVA), and perhaps provide some hypothesis in the discussion section.
2. Like mentioned above, have more discussions over the opposite conclusions from other work in other cancer types.

---

## Round 0.2 · accepted · Accept

All issues pointed by the reviewers were adequately addressed, and the revised manuscript is acceptable now.

·

Basic reporting

No comment

Experimental design

No comment

Validity of the findings

No comment

Additional comments

Almost all required revisions were done. Congratulations to your valuable work.

Reviewer 3 ·

Basic reporting

The authors have addressed my previous questions and no more comments.

Experimental design

The authors have addressed my questions. For my questions regarding stratification method used for immune cell infiltration, and also why 5.6 is used as a threshold, would recommend the authors to include the description that median value is used, in the methodology section.

Validity of the findings

The authors have addressed my previous questions and no more comments.